# Body Fat Reduction Effect of *Bifidobacterium breve* B-3: A Randomized, Double-Blind, Placebo Comparative Clinical Trial

**DOI:** 10.3390/nu15010028

**Published:** 2022-12-21

**Authors:** Hyun Kyung Sung, Sang Jun Youn, Yong Choi, Sang Won Eun, Seon Mi Shin

**Affiliations:** 1Department of Pediatrics, College of Korean Medicine, Semyung University, Jecheon 27136, Republic of Korea; 2RnBS Corp., Seoul 06032, Republic of Korea; 3Daehan Chemtech Co., Ltd., Seoul 01811, Republic of Korea; 4Department of Internal Medicine, College of Korean Medicine, Semyung University, Jecheon 27136, Republic of Korea

**Keywords:** *Bifidobacterium breve* B-3, placebo-controlled study, randomized trial, obesity, nutritional supplement

## Abstract

This double-blind, randomized clinical trial aimed to evaluate the efficacy and safety of *Bifidobacterium breve* B-3 (BB-3) for reducing body fat. Healthy individuals were randomized into the BB-3 or placebo group (1:1). Dual-energy X-ray absorptiometry was used to evaluate body fat reduction objectively. In the BB-3 group, body weight was lower than before BB-3 ingestion. Regarding waist circumference, hip circumference, and waist/hip circumference ratio, waist circumference and hip circumference were lower in the BB-3 group than in the placebo group at 12 weeks; the waist/hip circumference ratio was found to decrease at each visit in the BB-3 group, although there was no significant difference in the amount of change after 12 weeks. BB-3 did not cause any severe adverse reactions. Body fat was significantly lower in the BB-3 group than in the placebo group. In conclusion, ingesting BB-3 significantly reduces body weight, waist circumference, and hip circumference. Thus, BB-3 is safe and effective for reducing body fat.

## 1. Introduction

The recent obesity development theory is explained by the “carbohydrate-insulin model” (CIM) based on the hormonal response to highly processed carbohydrates rather than the “energy balance model” (EBM) theory, which posits that obesity occurs because energy intake is less than consumption [1,2]. In 2015, 107.7 and 603.7 million children and adults, respectively, were identified to be obese. The prevalence of obesity since 1980 has doubled in more than 70 countries and has continuously increased in most other countries [3]. From 1990 to 2017, the global deaths and disability-adjusted life years (DALYs) attributable to a high body mass index (BMI) have more than doubled for both females and males [4]. The Global Burden of Disease Study level 3 causes of DALYs associated with high BMI in 2017 were ischemic heart disease, stroke, diabetes mellitus, chronic kidney disease, hypertensive heart disease, and low back pain [4]. Obesity is also associated with inherent complications and various chronic diseases such as type 2 diabetes, high blood pressure, hyperlipidemia, arteriosclerosis, stroke, osteoarthritis, and obstructive sleep apnea. Obesity was reported to directly cause 80% of diabetes cases and 20% of heart disease cases worldwide [5,6]. Given the increasing prevalence of various cancers, a cure for obesity has gathered great interest [7,8]. The World Health Organization has recognized obesity as a global health problem that needs to be treated due to its increasing prevalence. Obesity is a complex disease mainly caused by excessive caloric intake and lack of exercise, although social, genetic, and environmental factors also affect its occurrence [9,10].

Recent studies have suggested that gut microbiota is a factor that may influence obesity. It has also been found to support the ability to regulate energy balance, fat storage, neurohormonal function, and the immune system [11,12,13,14]. Moreover, altering the composition of the gut microbial ecosystem has been proposed as a novel approach to treating obesity. This strategy mainly involves altering the composition of the gut microbiome in obese individuals by ingestion of beneficial microorganisms, i.e., probiotics [15]. When consumed as a functional food ingredient, probiotics have been recognized to help lactic acid bacterial growth, suppress harmful bacteria, and facilitate good bowel movement. They have also been recognized for their benefits in vaginal health, immunity regulation, intestinal health, skin protection against further ultraviolet damage, and skin moisture. Several previous studies have reported the effect of bifidobacteria on the improvement of weight or body fat-related indicators [16]. Among them, *Bifidobacterium breve* B-3 (BB-3) is a strain with a patent for its body-fat reduction action. A non-clinical test conducted on mice showed the efficacy of BB-3 against obesity [17]. 

This study aimed to confirm the effectiveness of BB-3 in reducing body fat in Koreans.

## 2. Materials and Methods

### 2.1. Study Design and Participants 

This randomized, double-blind, placebo comparative clinical trial (ver. 0.2, issue date 2021) recruited healthy individuals through a written notice posted on the hospital homepage and bulletin board of Semyung University Korean Medicine Hospital (Jecheon, Chungcheongbuk-do, Republic of Korea) until the target sample size was reached. Individuals willing to participate visited the Department of Internal Medicine and were screened according to the participant selection criteria presented in Table 1. The first participant was enrolled in April 2021. The total time for study participation was approximately 14 weeks, including a maximum of a 2-week wash-out period and safety assessment 2 weeks after the last visit. If a participant had a drug history of concomitant use of prohibited medication or food, a maximum 21-day wash-out period was required. 

Participants in the study were randomly classified into treatment and placebo groups on the second visit (within 3 weeks of the first visit), which served as the baseline time point.

The inclusion and exclusion criteria were rechecked before randomization, and participants who met the criteria were enrolled. A baseline assessment was performed, and 33-day supplies of the investigational product or placebo were provided to the participants. Follow-up visits occurred 28 (visit 3), 56 (visit 4), and 84 (visit 5) days after the baseline assessment (visit 2). In addition, a 5-day visit window was allowed. Vital signs, medical history/concomitant drug examinations, and efficacy and safety evaluations were performed during visits 3–5 (Figure 1). Laboratory and pregnancy tests were performed at visits 1 and 5. Participants were notified of the hospital visit schedules by the clinical trial investigator.

### 2.2. Intervention

The investigational product contained BB-3 as the main ingredient and 50 mg (11.11%) maltodextrin (85.89%), magnesium stearate (1.0%), and silicon dioxide (2.0%) at 450 mg/cap as excipients. The placebo drug contained maltodextrin (97.0%), magnesium stearate (1.0%), and silicon dioxide (2.0%) as excipients at 450 mg/cap. BB-3 contained 1 × 10^11^ colony-forming units (CFUs) of BB-3 per 1 g of corn starch, including BB-3 as the same raw material as the B-3-EX product sold commercially by Morinaga Dairy in Japan. The daily intake of BB-3 in the BB-3 group was 5 billion CFU/capsule/day (Appendix A). The test and placebo capsules were manufactured to be similar in shape, size, and color (Figure 2). The test capsules disintegrated in the stomach. Test or placebo drugs were ingested orally once daily for 12 weeks. Participants were prescribed 1 month’s dose at visit 2, visit 3, and visit 4 and were then encouraged to continue with the prescribed dose. The remaining unused capsules were returned at visit 2, visit 3, and visit 4 and counted to evaluate drug compliance.

The participants were guided to maintain their usual diet and exercise during the study period. However, they were banned from taking drugs or foods that could cause body fat loss. Drugs and food consumption, exercise activities, and diets followed before participation were allowed at the researcher’s discretion. Information about all medications, including the items or names, doses, and duration of the medications taken, was recorded at each visit. The intervention was interrupted under the following conditions: a serious adverse event, use of a drug or undergoing a physical procedure that could affect body fat and lipid levels, participants wanting to stop participating in the study, difficulties in the evaluation due to administrative reasons (e.g., violation of dosing method or visit schedule), and difficulties in follow-up due to participants’ personal reasons.

### 2.3. Randomization and Blinding

Stratified block randomization was performed. The participants were randomized to the placebo or experimental group in a 1:1 ratio. Using the SAS^®^ system’s randomization program, a random number sequence was created, starting with participant number 1. When packing food, the sponsor attached the food label for clinical trials according to the IP code list and supplied it to the test institution before the commencement of this clinical trial. The stratified block randomization method was used to prevent bias that could be involved in the allocation of intake groups, to increase comparability between groups, and to ensure balanced allocation. Stratified block randomization was performed at visit 2 according to sex (male and female). Randomization was performed using the web-based interactive web response system (IWRS), and the randomization code was reproducible by assigning a seed.

The randomization code and IP number were managed by a third-party individual unblinded to the data. The code and number were not disclosed until statistical analysis, except in cases where it was necessary to read the code owing to a serious medical emergency. The IP manager (or pharmacist) supplied the intervention for the clinical trial with an IP number assigned to the participant. In case of defect or damage to the intervention, another IP number was reassigned using the IWRS system to maintain the treatment arm. To maintain double blinding (all researchers and subjects participating in clinical trials), participant allocation details and serious adverse reactions and codes mentioned in the production, packaging, and labeling of products used in the clinical trial were sealed in an envelope by the person in charge of the trial. The code was not released until the end of the study, except in inevitable cases where the code needed to be checked. Clinical trial sponsors provided interventions that matched the registration number assigned to the selected participants.

### 2.4. Endpoints

The primary endpoint was the change in body fat mass (g) and body fat percentage (%) assessed on dual-energy X-ray absorptiometry (DEXA) at 12 weeks from baseline. The secondary endpoints were as follows: (1) changes in lean mass (g), body fat mass (g) by area (arms, legs, trunk, android, and gynoid), body fat percentage (%) by area (arms, legs, trunk, android, and gynoid), and lean mass (g) by area (arms, legs, trunk, android, and gynoid) assessed using DEXA at 12 weeks from baseline; (2) changes in total fat area, subcutaneous fat area, visceral fat area, and visceral fat area/subcutaneous fat area ratio measured using abdominal computed tomography (CT) 12 weeks from baseline; (3) changes in body weight, body mass index (BMI), waist circumference, hip circumference, waist/hip circumference ratio at 4, 8, and 12 weeks from baseline; (4) changes in blood lipid concentrations (total cholesterol, low-density lipoprotein (LDL)-cholesterol, triglyceride (TG), and high-density lipoprotein (HDL)-cholesterol), leptin, and adiponectin at 12 weeks from baseline. As the primary outcome measure, DEXA was used to assess the body fat mass and percentage at baseline and week 12. 

Total fat mass was measured in a supine position using a LUNAR Prodigy Vision scanner (software version 6.70; General Electric Medical Systems, Madison, WI, USA) and a whole-body DEXA scanner. Total fat mass and body fat mass were obtained using standard soft tissue measurement methods. Body fat mass was calculated as the amount of fat present in the section (chest, abdomen, and pelvis) surrounded by the virtual boundary line that separates the head and limbs when measured with a whole-body DEXA scanner. Abdominal CT was used to measure visceral fat area, subcutaneous fat area, total abdominal fat area, and visceral fat/subcutaneous fat area ratio at baseline and week 12. A CT scan was performed between the fourth and fifth lumbar vertebrae. The high accuracy of CT makes it the preferred method for measuring visceral and subcutaneous fat [18]. Body weight and BMI were measured at baseline and weeks 4, 8, and 12. Waist circumference, hip circumference, and waist circumference/hip circumference ratio were assessed at baseline and weeks 4, 8, and 12. Following the World Health Organization guidelines, waist circumference was measured at the midpoint between the lower margin of the last palpable rib in the midaxillary line and the top of the iliac crest. Hip circumference was measured at the largest circumference of the buttocks (World Health Organization. Waist circumference and waist-hip ratio: Report of a WHO expert consultation. Geneva, Switzerland: WHO Press; 2011). 

Serum lipid (total cholesterol, triglyceride, high-density lipoprotein cholesterol, and low-density lipoprotein cholesterol) and adipocytokine (leptin and adiponectin) concentrations were assessed at visit 1 and visit 5. Blood samples were collected after at least an 8-h fast and analyzed in the laboratory. Physical activity and dietary habits (24-h recall) were surveyed. Physical activity was evaluated at visits 2 and 5 using the Korean version of the short-form International Physical Activity Questionnaire (IPAQ). The IPAQ was selected as a questionnaire tool for various surveys conducted by the WHO; reliability validity studies were conducted in 12 countries and it is currently being used worldwide [19]. Through the IPAQ, participants were asked to recall and record the amount of activity for the week before visits 2 and 5. Dietary habits were analyzed using the 24-h recall method. A 24-h recall diary was prepared at visits 2 and 5. In this study, the participant’s diet was investigated through total calorie (kcal) analysis using the CAN-PRO program (CAN 5.0 Web ver.). 

### 2.5. Safety

Adverse reactions and side effects were evaluated through interviews during the visit or through blood and urine tests before and after the intervention.

### 2.6. Sample Size Calculation

We referred to the study by Cho et al. [20], who reported that the change in body fat mass of the herbal extract powder (Imperata cylindrical Beauvois, Citrus unshiu Markovich, and Evodia officinalis Dode) group was −1.6 kg, the change in body fat mass of the placebo group was −0.1 kg, and the unpaired *t*-test *p*-value between the two groups was 0.023. Based on these results, it was assumed that the effect size of this clinical trial was −1.5, and the pooled standard deviation was 2.2676 kg. Based on the change in body fat mass, which was calculated for the number of participants, the number of participants required to achieve a significance level of 5% and power of 84% was calculated to be 40 participants per group. Considering a dropout rate of 20%, 50 participants per group (=40/(1 − 0.2)) for a total of 100 participants were planned to be enrolled.

### 2.7. Statistical Analyses

A statistical hypothesis test was conducted at a two-sided significance level of 0.05. The study endpoints were analyzed using the number of participants, mean, and standard deviation. Furthermore, normally distributed data were analyzed using covariate analysis with baseline values and sex as the covariate. Non-normally distributed data were compared between groups using the Wilcoxon rank sum test. Continuous variables were reported as the mean and standard deviation and analyzed using the two-sample *t*-test or Wilcoxon rank sum test for inter-group comparison. Moreover, categorical variables were reported as frequencies and percentages and analyzed using Pearson’s chi-square test or Fisher’s exact test for inter-group comparison. Intra-group analysis was performed using paired *t*-tests or Wilcoxon signed-rank test. The per-protocol set was used to analyze the primary and secondary endpoints. The per-protocol set included all participants who completed the study protocol and had no major protocol deviations. The safety set included all participants who received at least one capsule of the investigational product and had at least one safety assessment; this set was used for the safety analysis. All statistical analyses were performed using SAS^®^ software (version 9.4, SAS, Cary, NC, USA).

## 3. Results

### 3.1. Participant Characteristics

The first participant was screened on 6 April 2021 and the last was screened on 30 July 2021. In total, 104 participants were evaluated; among them, 4 participants were excluded, and finally, 100 participants (51 participants in the BB-3 group and 49 participants in the placebo group) were enrolled. Consequently, 6 participants from each group (total: 12) dropped out, and thus, 83 participants (42 participants in the BB-3 group and 41 participants in the placebo group) completed the clinical trial (Figure 3). There was no significant between-group difference in age (46.55 ± 9.76 years in the BB-3 group vs. 45.02 ± 9.23 years in the placebo group, *p* = 0.3361). There were also no significant between-group differences in sex, height, weight, BMI, waist/hip circumference, fat mass index, fat-free mass index, and family history of obesity (Table 2).

### 3.2. Study Endpoints

The amount of body fat (g) was significantly lower after BB-3 intake than before (*p* = 0.0005). Meanwhile, in the placebo group, although body fat was lower after intake than before intake, the difference was not significant. Importantly, the amount of body fat was significantly lower in the BB-3 group than in the placebo group (*p* = 0.0170) (Table 3). The amount of body fat percentage (%) was decreased in both groups; however, the differences within and between groups were not significant (*p* = 0.3760) (Table 3). Regarding secondary outcomes, weight, BMI, waist circumference, and hip circumference measured at visits 2 and 5 were significantly lower than those measured at baseline values (Appendix A). In the BB-3 and placebo groups, there were no significant differences in the other secondary endpoints; however, total cholesterol, low-density lipoprotein (LDL)-cholesterol, triglycerides (TG), high-density lipoprotein (HDL)-cholesterol, and leptin, excluding adiponectin, showed a tendency to decrease after 12 weeks. Even so, there was no statistical significance between the two groups. (Appendix A).

### 3.3. Safety

In the BB-3 group, 15 (29.41%) participants had 21 adverse reactions; in the placebo group, 14 (28.57%) participants had 19 adverse reactions, with no significant between-group differences (*p* = 0.9262). The most common adverse reactions, such as muscle pain, headache, and injection site pain, occurred after COVID-19 vaccination. All other adverse reactions were mild and unrelated to the study drug and no serious adverse reactions occurred (Appendix A). For the hematological test results, the red blood cell, hemoglobin, hematocrit, and platelet levels before and after ingestion of BB-3 were significantly higher than before ingestion of food for human application in both the groups; however, both the groups showed changes near the normal range. There was no significant between-group difference in the amount of change (Appendix A). In the BB-3 group, the alanine aminotransferase (ALT) level was significantly higher after treatment than before, although the difference was not significant. There was also no significant between-group difference with respect to the amount of change in ALT levels (Appendix A). There were no significant changes in other laboratory parameters. There were also no significant normal/abnormal changes in the urine test results before/after treatment (Appendix A).

### 3.4. Physical Activity and Diet 

There were no significant between-group differences in the dietary intake before (0 weeks) and after (12 weeks) of the intervention. (Appendix A).

## 4. Discussion

To evaluate the body fat reduction effect of BB-3, a randomized, double-blind clinical trial was conducted in overweight adults in which BB-3 or a placebo was administered for 12 weeks. At 12 weeks, body weight and BMI were significantly lower in the BB-3 group than in the placebo group. Out of the waist circumference, hip circumference, and waist/hip circumference ratio, waist circumference and hip circumference were lower in the BB-3 group than in the placebo group at 12 weeks. 

The composition of the gut microbiota differs between lean and obese participants and is recognized as a therapeutic target of obesity. In a meta-analysis of randomized controlled trials to examine the effects of probiotic supplementation on body composition in overweight (BMI 25–30 kg/m^2^) and obese (BMI ≥ 30 kg/m^2^) participants, five studies reported changes in body fat percentage, and the pooled estimate showed that percent body fat was significantly lower in the intervention groups (−0.60%) than in the control groups, with low heterogeneity among the studies [21,22]. BB-3 is a promising anti-obesogenic strain. 

Some studies have reported a dose-dependent inhibition of body weight gain and visceral fat deposition and improved serum levels of total cholesterol, glucose, and insulin with BB-3 administration in diet-induced obese mice [23]. In humans, the daily intake of capsules containing a lyophilized powder of BB-3 at a dose of 5 × 10^10^ CFUs/day reduced body fat mass [16]. In addition, several clinical trials have reported the positive effects of probiotic strains on reducing visceral fat. The 12-week consumption of fermented milk containing *Lactobacillus gasseri* SBT2055 significantly reduced visceral fat areas in adults with accumulated visceral fat (81.2–178.5 cm^2^) [24]. Furthermore, the 12-week consumption of fermented milk containing *B. animalis* ssp. lactis GCL2505 was also recently reported to significantly reduce visceral fat areas in healthy participants with BMIs ranging between 23 kg/m^2^ and 30 kg/m^2^ [22,25]. 

In this clinical trial, the amount of body fat (g) assessed using DEXA, body weight, BMI, waist circumference, and hip circumference were significantly lower after BB-3 ingestion than before. Notably, similar to that in a previous study [22], visceral fat tended to be lower than baseline values in the BB-3 group, which seems to have affected the reduction in body fat in the android and trunk regions. In addition, although the reduction in body fat mass was effectively lower in the BB-3 group, the reduction in body fat percentage was not significant. However, the body fat percentage tended to decrease in the BB-3 group, particularly in the trunk and android areas. This is considered to be due to the decrease in visceral fat and indicates the body fat reduction effect of BB-3. Previous studies have shown that the probiotic strain *B. breve* B-3 increased the number of cells and the proportion of bifidobacteria in the intestine [22]. In addition, the upregulation of glucagon-like peptides and proglucagon expression, such as Fiaf in the intestine and adiponectin expression in the B-epiticular fat pad, has been shown to be effective in preventing obesity and insulin resistance [16].

Cani et al. [26] suggested that the regulation of intestinal peptides involved in the regulation of energy and glucose homeostasis could be one of the mechanisms involved in the improvement of the microbiota regulation of metabolic syndrome. They found that OFS administration increased colonic proglucagon, such as a glucagon-like peptide (GLP-2). GLP-1 and GLP-2 are produced and released by enteroendocrine L cells in the distal ileum and large intestine [27]. GLP-1 stimulates postprandial insulin secretion and reduces appetite by stimulating the hypothalamus and stomach. Studies of germ-free and normal mice have shown that the microbiota promotes the absorption of monosaccharides from the intestinal lumen and, consequently, induces new liver adipogenesis [28]. The fasting-inducing adipocyte factor, Fiaf, is a member of the angiopoietin-like family of proteins that are repressed in the intestinal epithelium. Fiaf is also a circulating lipoprotein lipase inhibitor and its inhibition is essential for microbial-induced triglyceride deposition in adipocytes [29]. In previous studies, Fiaf expression was significantly upregulated in the small intestine of B-supplemented mice [16]. BB-3 affects the mechanisms involved with the reduction in fat accumulation in adipocytes.

Obesity and insulin resistance are factors associated with metabolic syndrome. Hypertrophic adipocytes produce abnormal adipokines and cytokines, such as TNF-alpha, MCP-1, FFA, IL-6, and resistin, which inhibit insulin signaling in hepatocytes and induce insulin resistance. Meanwhile, adiponectin in normal adipocytes has been shown to play an important role in regulating energy homeostasis and insulin sensitivity. Studies in adiponectin transgenic mice suggest that insulin resistance is associated with increased expression of molecules involved in fatty acid oxidation (e.g., acyl-CoA oxidase), and molecules involved in energy dissipation (e.g., dissociation of proteins 2 and 3) have been shown to be related to increased fatty acid oxidation in skeletal muscle [30]. Adiponectin expression was significantly upregulated in the epididymal fat pad of *B. breve* B-3-fed mice. These results suggest that the level of adiponectin upregulated by *B. breve* B-3 administration is involved in improving insulin resistance by preventing adipocyte hypertrophy [16].

This current clinical trial found that body fat and body weight were lower in the BB-3 group than in the control group, consistent with the results of previous clinical trials. Overall, this result supports the assertion that BB-3 administration induces a decrease in visceral fat that results in a decrease in body fat mass. Previous experimental studies showed that BB-3 intake affects fat cells or fat metabolism and has a body fat reduction effect. However, improvements in insulin resistance and related indicators still need to be investigated. This current clinical trial provides baseline evidence for future studies on the body fat reduction effect of BB-3 in humans. In this clinical trial, in the BB-3 group, there was a statistically significant decrease in body fat mass as well as a decrease in lean body mass, although not statistically significant. This decrease in body fat mass and lean mass produced a statistically significant weight loss effect. On the contrary, there was no significant decrease in body fat percentage. This is because, even though there was a statistically significant decrease in body fat mass, the decrease in body fat percentage was offset by the decrease in lean mass. For reference, the body fat percentage is composed of the ratio of body fat mass to muscle mass, and most of the lean mass is composed of muscle mass. Diet programs and functional food intake reduce both fat mass and lean body mass, resulting in weight loss. However, it is important to prevent a decrease in lean mass because it causes the weight to increase again (the yo-yo effect) [31]. Therefore, in order to maintain weight loss in the long term, it is important to incorporate appropriate exercises [32].

In this clinical trial, lifestyle factors, such as exercise and eating habits, were adjusted to remain the same during the test period and no special exercise prescription was administered. Intake of BB-3 resulted in a decrease in lean body mass along with a decrease in body fat mass that can occur with initial weight loss. This resulted in a statistically non-significant decrease in body fat percentage. In the future, if a long-term study that combines exercise prescription with the intake of BB-3 is conducted, it can be assumed that the effect of weight loss due to the reduction in body fat mass and percentage of body fat, without reduction in lean body mass, will be confirmed. The limitation of this study was that it was a single-center study and not a crossover study. Multicenter and crossover studies in the future will generate clearer and more reliable evidence. In addition, studies that can confirm changes in intestinal microbes after taking BB-3 and mechanistic studies on the effect of BB-3 on body fat reduction are considered necessary.

## 5. Conclusions

Body fat mass (g) was significantly lower after BB-3 intake. Meanwhile, although body fat percentage (%) and fat-free mass (g) also decreased, no significant changes were observed. The reduction in body fat mass was particularly affected by the reduction in body fat mass in the trunk and android regions. Body weight and BMI were lower after the intervention than before in the BB-3 group, with parameters showing a continuous decrease at every visit (weeks 4, 8, and 12). In the 12th week, body weight and BMI were significantly lower in the BB-3 group than in the placebo group. Although body fat mass was effectively reduced in the BB-3 group, there was no statistically significant change in body fat percentage in the placebo group. Waist- and hip-circumferences were significantly lower in the BB-3 group than in the placebo group at 12 weeks. The waist/hip circumference ratio also decreased at each visit in the BB-3 group but no significant change was observed at 12 weeks in the placebo group. Collectively, these findings indicate that BB-3 can safely and effectively reduce not only body fat but also body weight, waist circumference, and hip circumference. Future research should be conducted on the effects of BB-3 on the intestinal environment, its mechanisms, and the effects of hormone changes involved in human metabolism.

## Figures and Tables

**Figure 1 nutrients-15-00028-f001:**
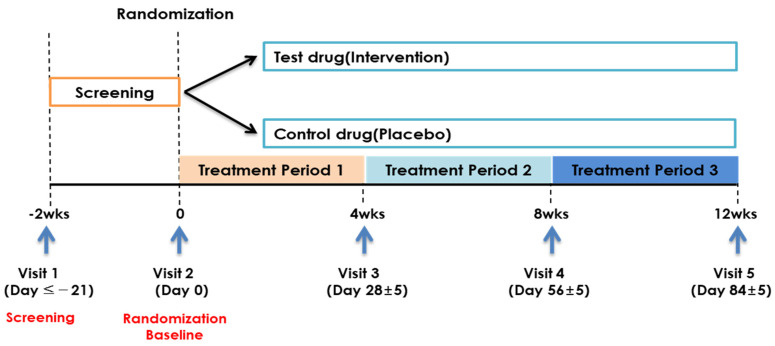
Clinical trial timeline.

**Figure 2 nutrients-15-00028-f002:**
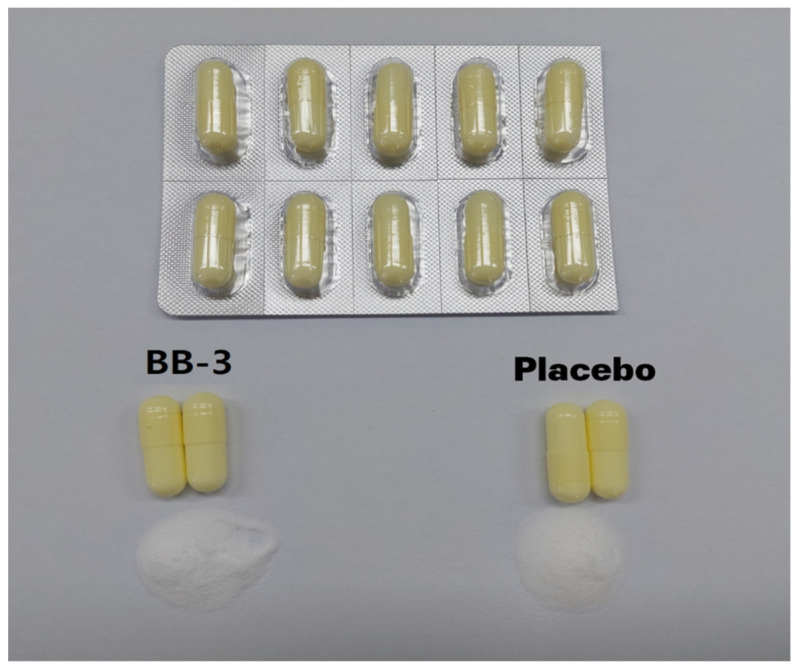
BB-3 and placebo capsules. BB-3: *Bifidobacterium breve* B-3.

**Figure 3 nutrients-15-00028-f003:**
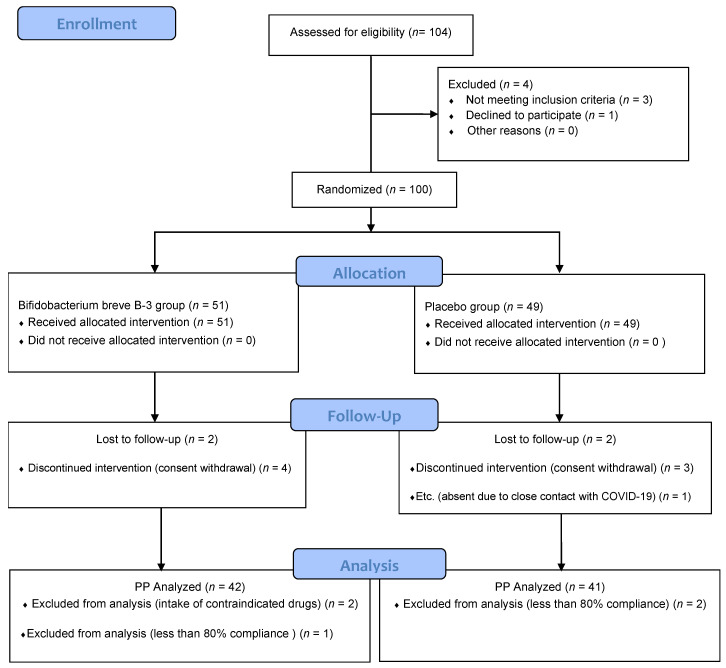
Trial flowchart. COVID-19: coronavirus disease 2019, PP: per protocol.

**Table 1 nutrients-15-00028-t001:** Participant selection criteria.

**Inclusion Criteria**
(1) Age 19–60 years
(2) Body mass index (BMI) of ≥25 kg/m^2^ and <30 kg/m^2^
(3) Able to provide written informed consent
**Exclusion Criteria**
(1) Severe cerebrovascular disease (cerebral infarction and cerebral hemorrhage), heart disease (angina pectoris, myocardial infarction, heart failure, and arrhythmia requiring treatment), or malignant tumor within the last six months. However, participants with a medical history of cerebrovascular disease or heart disease who were clinically stable could participate in the trial at the investigator’s discretion
(2) Taking drugs that affected body weight (fat absorption inhibitors and appetite suppressants, health food/supplements related to obesity, psychiatric drugs such as depression, beta-blockers, diuretics, contraceptives, steroids, and female hormones) within the last month
(3) Obese or overweight due to endocrine diseases such as hypothyroidism and Cushing’s syndrome
(4) Maintenance treatment for gastrointestinal disorders (gastric ulcer, chronic digestive disorder, and irritable bowel syndrome)
(5) Psychologically significant medical history or current disease (schizophrenia, epilepsy, anorexia, and bulimia) or a history of alcohol and other drug abuse
(6) Judgment of inability to exercise due to musculoskeletal disorders
(7) Fasting blood sugar of ≥126 mg/dL, random blood sugar of ≥200 mg/dL, or patients with diabetes taking oral hypoglycemic agents or insulin
(8) Uncontrolled hypertension (blood pressure >160/100 mmHg measured after a 10-min rest)
(9) Alanine aminotransferase(AST) or Alkaline phosphatase(ALT) level at least 2.5 times higher than the laboratory’s upper limit of normal
(10) Creatinine levels more than twice the upper limit of normal in the testing institute
(11) Weight loss ≥5% within the last three months
(12) Participation in a commercial obesity program within the last three months
(13) Participation in an obesity clinical trial within the last six months
(14) Pregnancy, lactation, or was planning to become pregnant during the study period
(15) An allergic reaction to the food study drug
(16) Others were considered unsuitable for the study at the discretion of the principal investigator
(17) The intake of probiotics within the last month

**Table 2 nutrients-15-00028-t002:** Participant characteristics by group.

		BB-3 Group (*n* = 42)	Placebo Group (*n* = 41)	*p*-Value
Sex	Male	15 (35.71)	11 (26.83)	0.3829 ^1^
	Female	27 (64.29)	30 (73.17)	
Age, years		46.55 ± 9.76	45.02 ± 9.23	0.3361 ^2^
Height, cm		164.17 ± 9.15	162.22 ± 8.87	0.3343 ^2^
Weight, kg		72.71 ± 8.22	70.82 ± 8.17	0.2948 ^3^
Body mass index		26.93 ± 1.29	26.85 ± 1.38	0.7325 ^2^
Waist circumference, cm		88.39 ± 4.58	87.62 ± 5.55	0.4928 ^3^
Hip circumference, cm		99.04 ± 3.66	98.60 ± 3.89	0.6003 ^3^
Fat mass index, g		25,446.90 ± 4405.20	26,163.00 ± 3809.42	0.4311 ^3^
Fat-free mass index, g		47,094.50 ± 8707.45	44,334.07 ± 8532.53	0.0965 ^2^
Family history of obesity	Yes No	16 (38.10) 26 (61.90)	16 (39.02) 25 (60.98)	0.9307 ^1^

^1^ *p*-value for the chi-square test, ^2^ *p*-value for the Wilcoxon rank sum test, ^3^ *p*-value for the two-sample *t*-tests. Data are presented as *n* (%) or as the mean ± SD. Abbreviations: BB-3: *Bifidobacterium breve* B-3, SD: standard deviation.

**Table 3 nutrients-15-00028-t003:** DEXA results at 12 weeks from baseline according to the groups (PPS).

		BB-3 Group (*n* = 42)	Placebo Group (*n* = 41)
Body fat mass, g	V2	25,446.90 ± 4405.20	26,163.00 ± 3809.42
	V5	24,859.86 ± 4382.83	26,098.63 ± 4022.56
	V5-V2	−587.05 ± 1004.42	−64.37 ± 933.76
	*p*-value	0.0005 ^1^	0.6613 ^1^
	Difference V5-V2 (Tx-Px)	−522.68 ± 970.17	
	LS mean difference ^5^	−528.56	
	*p*-value	0.0170 ^3^	
Body fat percentage (%)	V2	36.60 ± 6.67	38.73 ± 6.14
	V5	36.28 ± 6.77	38.64 ± 6.25
	V5-V2	−0.32 ± 1.26	−0.09 ± 0.97
	*p*-value	0.1097 ^1^	0.5431 ^1^
	Difference V5-V2 (Tx-Px)	−0.22 ± 1.12	
	LS mean difference ^5^	−0.23	
	*p*-value	0.3760 ^3^	
Fat-free mass, g	V2	47,094.50 ± 8707.45	44,334.07 ± 8532.53
	V5	46,622.79 ± 8539.42	44,362.80 ± 8454.12
	V5-V2	−471.71 ± 1500.65	28.73 ± 840.49
	*p*-value	0.0916 ^2^	0.8279 ^1^
	Difference V5-V2 (Tx-Px)	−500.45 ± 1220.14	
	*p*-value	0.1172 ^4^	

^1^ *p*-value for the paired *t*-tests, ^2^*p*-value for the Wilcoxon signed rank test, ^3^ *p*-value for ANCOVA adjusted for baseline values and sex, ^4^ *p*-value for the Wilcoxon rank sum test, ^5^ ANCOVA results adjusted for baseline values and sex. Tx: BB-3, Px: placebo, ANCOVA: analysis of covariance, PPS: per protocol set. Data are presented as the mean ± SD. Abbreviations: V2: Visit 2, V5: Visit 5.

## Data Availability

The data presented in this study are included in the article/Appendix A. Further inquiries can be directed to the corresponding author.

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
