# Peer review of "Body Fat Reduction Effect of *Bifidobacterium breve* B-3: A Randomized, Double-Blind, Placebo Comparative Clinical Trial"

_nutrients, 2022, doi:10.3390/nu15010028_

Round 1
Reviewer 1 Report
Introduction Line 35: please cite ref
Methods: Line 77 what are prohibited diet and medications?
Patients on steroids included or excluded. ?
What is the rationale for analyzing secondary objectives by area?
Table 3: V2 and V5 - please mention in the foot tables as visit
Table 3 values does appear to be skewed data?
Supplementary tables are missing couldnt verify the results.
Authors are saying the body fat mass decreased and weight decreased but not the fat percentage so could it be related to overall proportionate drop in weight? In that case whether this is actually a direct effect from the intervention?
Author Response
Response to Reviewer 1 Comments
We thank you for your thoughtful suggestions and insights. The manuscript has benefited from these insightful suggestions.
The manuscript has been rechecked and the necessary changes have been made in accordance with the your suggestions. The responses to all comments have been prepared and attached herewith.
Thank you for your consideration
Point 1: Introduction Line 35: please cite ref
Response 1: Added Reference 4. Thank you for your careful review.
Point 2: Methods: Line 77 what are prohibited diet and medications?
Response 2: You can find out in Exclusion criteria 2) in Table 1.
2) Taking drugs that affected body weight (fat absorption inhibitors and appetite suppressants, health food/supplements related to obesity, psychiatric drugs such as depression, beta blockers, diuretics, contraceptives, steroids, and female hormones) within the last 1 month
Point 3:Patients on steroids included or excluded. ?
Response 3: Yes, It was mentioned as follows : Exclusion criteria 2) in Table 1.
2) Taking drugs that affected body weight (fat absorption inhibitors and appetite suppressants, health food/supplements related to obesity, psychiatric drugs such as depression, beta blockers, diuretics, contraceptives, steroids, and female hormones) within the last 1 month
Point 4: What is the rationale for analyzing secondary objectives by area?
Response 4: In taking BB-3, an evaluation was conducted on the decrease in each area to clearly determine whether it was a decrease in body fat in the trunk or extremities such as the lower limbs. This content has not been added to the manuscript.
Point 5: Table 3: V2 and V5 - please mention in the foot tables as visit
Response 5: It was mentioned as follows. Abbreviations: V2: Visit2, V5: Visit5
Point 6: Table 3 values does appear to be skewed data?
Response 6: The values in the table 3 are not normally distributed. So We used the Wilcoxon rank sum test which is a non-parametric statistical method.
Point 7: Supplementary tables are missing couldnt verify the results.
Response 7: I uploaded the supplementary table as a file, but there must have been an error. I will upload it again. We apologize for any inconvenience you may have experienced while reviewing your dissertation.
Point 8: Authors are saying the body fat mass decreased and weight decreased but not the fat percentage so could it be related to overall proportionate drop in weight? In that case whether this is actually a direct effect from the intervention?
Response 8: Body fat mass and body fat percentage decreased in both the BB-3 group and the placebo group, and the range of change was larger in the BB-3 group. In addition, there was statistical significance after 12 weeks only in the BB-3 group in body fat mass ex-cluding lean body mass and body fat percentage. This is thought to be because the de-crease in lean mass was large in the BB- group. Changes in energy balance following weight loss are related to compensatory changes in energy expenditure, namely reductions in total energy expenditure, energy expenditure at rest, and energy expenditure by physical activity. The reduction in total energy expenditure is a result of a decrease in body mass (the reduction in lean mass due to weight loss is 14-23% of the weight loss in case of diet control, and the degree of loss is mitigated by exercise)[33], as well as metabolic. It is also related to an increase in efficiency. Lower measured resting en-ergy expenditure than predicted by reduced fat mass and fat free mass (reduction of 15 kcal per 1 kg of weight loss)[33] was also consistent with maintaining reduced body weight. It is the result of metabolic adaptation[34,35]. Therefore, in obese subjects, changes in energy balance due to a decrease in fat mass accompanying weight loss in-crease the possibility of weight gain again, so it is necessary to continuously reduce en-ergy intake and increase energy consumption by increasing physical activity after weight loss.( Kayoung Lee, Long-term Weight Loss Maintenance Korean J Obes 2015 December;24(4):179-183 http://dx.doi.org/10.7570/kjo.2015.24.4.179). Based on this point, if BB-3 intake, which suppresses fat accumulation and improves insulin resistance, and exercise to increase energy consumption are combined, it will be possible to induce a longer-term weight loss effect.

Reviewer 2 Report
In the manuscript “Anti-obesity effect of Bifidobacterium breve B-3: A randomized, double-blind, placebo comparative clinical trial” by Sung et al., it evaluates the efficacy and safety of the probiotic BB-3 for reducing body fat in healthy individuals in a double-blind, randomized clinical trial. There are many major concerns about this study:
Title: the term “anti-obesity” effect does not represent what is shown in the manuscript. Title needs to be changed
Exclusion criteria:
Was consumption of probiotics as part of the exclusion criteria?
Item #3 excludes Obese or overweight due to endocrine diseases such as hypothyroidism and Cushing’s syndrome. But in the discussion, it is mentioned that this double-blind clinical trial was conducted in overweight adults (Line 277) and overweight is defined by a BMI of ≥25 – 29.9 kg/m2 (Line 285). However, the inclusion criteria allow a BMI of ≥25 kg/m2 and <30 kg/m2 Please clarify
Authors mentioned that they measured changes in blood lipid concentrations (total cholesterol, low-density lipoprotein [LDL]-cholesterol, triglyceride [TG], and high-density lipoprotein [HDL]-cholesterol), leptin, and adiponectin at 12 weeks from baseline. But results of this analysis are not mentioned in the manuscript.
Line 227. 88 participants (45 participants in the BB-3 group and 43 participants in the placebo group) completed the clinical trial (Fig 3). But table 2 has 42 participants for BB3 and 41 for placebo. Check numbers
The data are not particularly clearly presented. The p-values show in table3 are not easy to follow. For example, comparison of visit 2 between BB3 and placebo group does not have a defined p-value.
The discussion is difficult to follow and talks mostly about previous studies. Authors tried to explain the mechanism behind BB3 to suppress body weight gain and visceral fat by using data from previous studies, but the data reported in the manuscript is not sufficient to support their hypothesis.
Author Response
Response to Reviewer 2 Comments
We thank you for your thoughtful suggestions and insights. The manuscript has benefited from these insightful suggestions.
The manuscript has been rechecked and the necessary changes have been made in accordance with the your suggestions. The responses to all comments have been prepared and attached herewith.
Thank you for your consideration
Point 1: In the manuscript “Anti-obesity effect of Bifidobacterium breve B-3: A randomized, double-blind, placebo comparative clinical trial” by Sung et al., it evaluates the efficacy and safety of the probiotic BB-3 for reducing body fat in healthy individuals in a double-blind, randomized clinical trial. There are many major concerns about this study:
Response 1: As you pointed out, I made the following corrections: The meaning of the title has been reduced to the body fat reduction effect, not the anti-obesity effect. In addition, the effect of BB-3 intake on body fat loss was further added to the review. In other words, both the BB-3 group and the control group tended to decrease the body fat mass and body fat percentage, and the range of change was larger in the BB-3 group. However, only the body fat mass showed a difference between the two groups before and after taking it. This is considered to be because the decrease in lean mass was large in the BB- group. This suggests that in obese subjects, a decrease in lean mass accompanied by weight loss appears at the beginning of weight loss, and activities to increase energy consumption, such as exercise, are required for continued weight loss.
Point 2: Title: the term “anti-obesity” effect does not represent what is shown in the manuscript. Title needs to be changed
Response 2: The title of this study was modified from anti-obesity effect to body fat reduction effect.
In addition, the content of the study was modified to have a body fat reduction effect, not an anti-obesity effect. Thank you for your review.
Exclusion criteria:
Point 3: Was consumption of probiotics as part of the exclusion criteria?
Response 3: Yes, probiotics are excluded. Added content to table 1.
17) Participations taking probiotics within 1 month
Point 4: Item #3 excludes Obese or overweight due to endocrine diseases such as hypothyroidism and Cushing’s syndrome. But in the discussion, it is mentioned that this double-blind clinical trial was conducted in overweight adults (Line 277) and overweight is defined by a BMI of ≥25 – 29.9 kg/m2 (Line 285). However, the inclusion criteria allow a BMI of ≥25 kg/m2 and <30 kg/m2 Please clarify
Response 4: Overweight is defined as a BMI of ≥25 – 29.9 kg/m2, which is the same as the inclusion criteria of a BMI of ≥25 kg/m2 and <30 kg/m2. This means that a BMI of 29.9 kg/m2 can participate in a clinical trial, but a BMI of 30 kg/m2 cannot.
Point 5: Authors mentioned that they measured changes in blood lipid concentrations (total cholesterol, low-density lipoprotein [LDL]-cholesterol, triglyceride [TG], and high-density lipoprotein [HDL]-cholesterol), leptin, and adiponectin at 12 weeks from baseline. But results of this analysis are not mentioned in the manuscript.
Response 5: As pointed out, the results blood lipid concentrations are described as follows.
Total cholesterol, low-density lipoprotein [LDL]-cholesterol, triglycerides [TG], and high-density lipoprotein [HDL]-cholesterol), and leptin, excluding adiponectin, showed a tendency to decrease after 12 weeks. However, there was no statistical significance between the two groups
Point 6: Line 227. 88 participants (45 participants in the BB-3 group and 43 participants in the placebo group) completed the clinical trial (Fig 3). But table 2 has 42 participants for BB3 and 41 for placebo. Check numbers
Response 6: It has been modified as follows. Consequently, 6 participants from each group (total: 12) dropped out, and thus, 83 participants (42 participants in the BB-3 group and 41 participants in the placebo group) completed the clinical trial (Fig 3).
Point 7: The data are not particularly clearly presented. The p-values show in table3 are not easy to follow. For example, comparison of visit 2 between BB3 and placebo group does not have a defined p-value.
Response 7: In the table below, there are no statistically significant differences between the baseline values in the two groups.
Table. DEXA results(Body fat mass, Body fat percentage, Fat-free mass) at baseline(V2)
|
Variable |
Visit |
BB-3 group(N=42) |
Placebo group(N=41) |
p-value |
|
Body fat mass |
Baseline(V2) |
25446.90±4405.20 |
26163.00±3809.42 |
0.43111) |
|
Body fat percentage |
Baseline(V2) |
36.60±6.67 |
38.73±6.14 |
0.10992) |
|
Fat-free mass |
Baseline(V2) |
47094.50±8707.45 |
44334.07±8532.53 |
0.09652) |
1) Unpaired T-test, 2) Wilcoxon rank sum test
The EMEA(CHMP) guideline “Guideline on adjustment for baseline covariates in clinical trials” includes the following:
7.2. Baseline comparison
Statistical testing for baseline imbalance has no role in a trial where the handling of the randomisation and blinding has been fully satisfactory. Baseline summaries with respect to the main covariates should be presented and discussed from a clinical point of view as any observed imbalance will be a random phenomenon.
Comparing the demographic characteristics between the two groups in Table 2, randomization was performed well in this clinical trial.
Moreover, there were no evidence to suggest that there was a statistically significant or clinically significant difference between the baseline values in the two groups.
Therefore, we do not present p-values for comparison of baseline values in the table according to the EMEA(CHMP) guideline but summarized the statistics and p-values for comparing the difference between the two group.
Point 8: The discussion is difficult to follow and talks mostly about previous studies. Authors tried to explain the mechanism behind BB3 to suppress body weight gain and visceral fat by using data from previous studies, but the data reported in the manuscript is not sufficient to support their hypothesis.
Response8: Body fat mass and body fat percentage decreased in both the BB-3 group and the placebo group, and the range of change was larger in the BB-3 group. In addition, there was statistical significance after 12 weeks only in the BB-3 group in body fat mass ex-cluding lean body mass and body fat percentage. This is thought to be because the de-crease in lean mass was large in the BB- group. Changes in energy balance following weight loss are related to compensatory changes in energy expenditure, namely reductions in total energy expenditure, energy expenditure at rest, and energy expenditure by physical activity. The reduction in total energy expenditure is a result of a decrease in body mass (the reduction in lean mass due to weight loss is 14-23% of the weight loss in case of diet control, and the degree of loss is mitigated by exercise)[33], as well as metabolic. It is also related to an increase in efficiency. Lower measured resting en-ergy expenditure than predicted by reduced fat mass and fat free mass (reduction of 15 kcal per 1 kg of weight loss)[33] was also consistent with maintaining reduced body weight. It is the result of metabolic adaptation[34,35]. Therefore, in obese subjects, changes in energy balance due to a decrease in fat mass accompanying weight loss in-crease the possibility of weight gain again, so it is necessary to continuously reduce en-ergy intake and increase energy consumption by increasing physical activity after weight loss.( Kayoung Lee, Long-term Weight Loss Maintenance Korean J Obes 2015 December;24(4):179-183 http://dx.doi.org/10.7570/kjo.2015.24.4.179). Based on this point, if BB-3 intake, which suppresses fat accumulation and improves insulin resistance, and exercise to increase energy consumption are combined, it will be possible to induce a longer-term weight loss effect.

Round 2
Reviewer 1 Report
Table 3: Authors acknowledged the data are not normally distributed. Why T test was used in the Body fat mass and body fat % and for the lean mass used Wilcoxon test.
I fail to understand authors reply to point 8
Point 8: Authors are saying the body fat mass decreased and weight decreased but not the fat percentage so could it be related to overall proportionate drop in weight? In that case whether this is actually a direct effect from the intervention?
Authors have noted that " In addition, there was statistical significance after 12 weeks only in the BB-3 group in body fat mass ex-cluding lean body mass and body fat percentage. This is thought to be because the de-crease in lean mass was large in the BB- group. "
Is lean body mass decreased in the BB group? The decrease was not significant. If authors say that weight is decreased, lean body mass decreased and body fat mass is decreased as well while body fat percentage is unchanged, then won't it imply that the weight reduction properties of BB is not specific to fat proportion?
Author Response
Thank you for giving me the opportunity to submit a revised draft of my manuscript titled “Body fat reduction effect of Bifidobacterium breve B-3: A randomized, double-blind, placebo comparative clinical trial” to MDPI Nutrients Special Issue "Effects of Probiotics on the Human Metabolome". I appreciate the time and effort dedicated to providing your valuable feedback on this manuscript. I have been able to incorporate changes to reflect most of the suggestions provided by the reviewers. I have marked the changes within the manuscript in red font.
Here is a point-by-point response to the reviewers’ comments and concerns.
Please see the attachment

Reviewer 2 Report
Comments:
Change "participations" in Table 1. Participant selection criteria.
Line 39. use italics for B. breve. Same for lines 338 and 339, reference 22, 23, 24, 25, 26, 27.
Line 292. 50 CFUs/day. Is this value correct?
Due to the high variability of the gut microbiota composition among individuals. Did you observe a difference among participants in the BB-3 group in response to the treatment for the variables you measured, specially the ones that significantly changed at the last time point? Did you see responders and non-responders in the BB-3 group? Also, did the exercise frequency and dietary habits correlated with the response to BB-3 for decreasing boy fat and BMI?
Please explain in more detail results of Table S8. What does it mean a 2401.50 value? What are the units? Is this a high or a low value?
Minami et al. 2018 reported that triglyceride levels decreased and HDL-cholestero improved the BB-3 group. Which could be the explanation that in this case you did not observe changes in blood lipid levels?
Author Response
Thank you for giving me the opportunity to submit a revised draft of my manuscript titled “Body fat reduction effect of Bifidobacterium breve B-3: A randomized, double-blind, placebo comparative clinical trial” to MDPI Nutrients Special Issue "Effects of Probiotics on the Human Metabolome". I appreciate the time and effort dedicated to providing your valuable feedback on this manuscript. I have been able to incorporate changes to reflect most of the suggestions provided by the reviewers. I have marked the changes within the manuscript in red font.
Here is a point-by-point response to the reviewers’ comments and concerns.
Point 1: Change "participations" in Table 1. Participant selection criteria.
Response #1: Thank you for the suggestion. I have modified it.
Point 2: Line 39. use italics for B. breve. Same for lines 338 and 339, reference 22, 23, 24, 25, 26, 27.
Response #2: Thank you for your careful review. We italicized it throughout the manuscript.
Point 3: Line 292. 50 CFUs/day. Is this value correct?
Response #3: We apologize, as this was an incorrect description. We revised the section as follows:
In humans, the daily intake of capsules containing a lyophilized powder of BB-3 at a dose of 5 × 1010 CFUs/day reduced body fat mass (Line 300)
Point 4: Due to the high variability of the gut microbiota composition among individuals. Did you observe a difference among participants in the BB-3 group in response to the treatment for the variables you measured, specially the ones that significantly changed at the last time point?
Did you see responders and non-responders in the BB-3 group?
Also, did the exercise frequency and dietary habits correlated with the response to BB-3 for decreasing body fat and BMI?
Response #4: Through experimental studies and previous clinical studies, it is thought that taking BB-3 affects changes in the intestinal microflora of subjects. However, we believe that this is a limitation of this study because it was impossible to confirm changes in intestinal microflora (Lines 378-380). When the research team conducted this clinical trial, all subjects were asked questions about the amount of physical activity and dietary intake, and all subjects responded. In addition, the research team was instructed not to change any lifestyle habits that could affect weight loss during the clinical trial period and to maintain the same habits as before participating in the clinical trial (line 112). Regarding food intake, participants were asked about the amount of food consumed 1 day before the visit, and the IPAQ was used to determine the amount of activity 1 week before the visit. Therefore, it cannot be seen as an absolute reflection of the amount of food intake and activity during the entire clinical trial period. It is thought that the intervention of taking BB-3 affected body fat loss and weight loss.
Point 5: Please explain in more detail results of Table S8. What does it mean a 2401.50 value? What are the units? Is this a high or a low value?
Response #5: The International Physical Activity Questionnaire (IPAQ) was used to assess physical activity, while the 24-hour dietary recall method was used to assess dietary habits. (Lines 184-185)
The IPAQ is used to calculate the total amount of physical activity and total energy expenditure, and the unit is expressed as MET-min/week and kcal/week, respectively.
2401.50 is lower than 2848.59, and 2401.50 MET-min/week means an activity equivalent to 3065.91 kcal/week (V2 activity in the BB-group). In the BB-3 group, activity increased after 12 weeks compared to that in the placebo group. However, it was not statistically significant. Food intake decreased more in the placebo group after 12 weeks than in the BB-group, but it was not statistically significant.
This is described in line 278.
Point 6: Minami et al. 2018 reported that triglyceride levels decreased and HDL-cholesterol improved the BB-3 group. Which could be the explanation that in this case you did not observe changes in blood lipid levels?
Response #6: Although HDL cholesterol decreased, the difference between the two groups for the change in HDL cholesterol after 12 weeks was not statistically significant. As a result, taking BB-3 has been shown to have a positive effect on improving lipid levels. In addition, the lipid levels of all participants in this study were within the normal range, and the lipid levels tended to decrease.
